# Revisiting the relative effectiveness of slaughterhouses in Ireland to detect tuberculosis lesions in cattle (2014–2018)

**Rischi Robinson Male Here**[1]*, **Eoin Ryan**[2], **Philip Breslin**[2], **Klaas Frankena**[1], **Andrew William Byrne**[3]

**1** Adaptation Physiology Group, Wageningen University & Research, Wageningen, The Netherlands, **2** Ruminant Animal Health Division, Department of Agriculture, Food and the Marine, Backweston Co., Kildare, Ireland, **3** Department of Agriculture, Food and the Marine, One-Health Scientific Support Unit, National Disease Control Centre (NDCC), Agriculture House, Dublin 2, Ireland

* rischi.malehere@wur.nl, robinsonmalher@gmail.com

**Data Availability Statement:** The data used in this analysis are held by the Department of Agriculture Food and the Marine (DAFM). Research access to these data will be considered on an individual

## Abstract

Slaughterhouse or meat factory surveillance to detect factory lesions (FL) at slaughter is an important part of the bovine tuberculosis (bTB) eradication program in Ireland. The objective of this study was to quantify the effectiveness of Irish slaughterhouses or factories in submitting FL and the proportion of those submitted FL confirmed as being due to bTB in slaughtered cattle, and to identify and quantify the association of risk factors at animal, herd, and factory level with FL submission and confirmation. The data consisted of 6,611,854 animals slaughtered in Irish factories from 2014 to 2018 obtained from the Department of Agriculture, Food and Marine (DAFM), Ireland. Selected risk factors for this study included factory, year and month of slaughter, age, sex, breed, animal movement, ever inconclusive in the standard or severe skin test, herd type, herd size, and bTB history. The association of each risk factor on the FL submission and confirmation risk were analysed with univariable followed by a multivariable logistic regression with herd as random effect. Factories were ranked and compared based on the odds ratio (OR) obtained from the univariable (crude OR) and multivariable (adjusted OR) analysis. The average submission risk of all factories was 20 per 10,000 animals slaughtered, ranging from 1 to 42 per 10,000 animals slaughtered, and the average confirmation risk over all factories was 40.72%, ranging from 0.00 to 61.84%. The odds of submitting and confirming FL as bTB positive were higher in animals over eight years old compared to animals 1–2 years old (OR = 1.91, 95 CI 95% 1.77–2.06 and OR = 4.05, 95% CI 3.17–5.18, respectively), and were higher in animals that ever had inconclusive skin result based on severe interpretation (OR = 2.83, 95% CI 2.44–3.27 and OR = 4.48, 95% CI 2.66–7.54, respectively), animals originating from sucklers herds (OR = 1.08, 95% CI 1.02–1.14 and OR = 1.31, 95% CI 1.11–1.55, respectively), or herds with bTB history in the past three years (OR = 4.46, 95% CI 4.28–4.66 and OR = 319.90, 95% CI 237.98–430.04, respectively). The odds of FL submission and confirmation decreased as the herd size increased (OR = 0.95, 95% CI 0.93–0.96 and OR = 0.82, 95% CI 0.78–0.86, respectively). An inverse relationship of FL submission and confirmation was present for variable sex and inconclusive skin result with standard interpretation, where submission

request basis. Requests should be directed to ERAD division at DAFM (ERAD@agriculture.gov. ie).

**Funding:** The article processing costs associated with the publication of this study were provided by the Department of Agriculture, Food and Marine, Ireland. RRMH was sponsored by Indonesia Endowment Fund for Education scholarship (LPDP, https://lpdp.kemenkeu.go.id) from the Ministry of Finance, the Republic of Indonesia. The funder had no role in study design, data collection, and analysis, decision to publish, or preparation of the manuscript.

**Competing interests:** The authors have declared that no competing interests exist.

odds were higher in males (OR = 1.05, 95% CI 1.00–1.10) and ever inconclusive animals (OR = 74.24, 95% CI 69.39–79.43), although the confirmation odds were lower (males OR = 0.66, 95% CI 0.56–0.76; ever inconclusive OR = 0.44, 95% CI 0.36–0.54). The crude and adjusted ranking of factories did not differ greatly for FL submission, indicating that factory-related factors may contribute significantly to the submission variation between factories. However, a substantial difference between crude and adjusted confirmation ranking was present which may indicate that animal and herd-related factors were associated to variation in confirmation risk between factories.

## Introduction

Bovine tuberculosis (bTB) is a chronic infectious disease caused by the *Mycobacterium tuberculosis* complex, especially *Mycobacterium bovis*, affecting cattle, wildlife, and humans worldwide. In industrialised countries, *M. bovis* infection has less zoonotic impact compared to developing countries [1] but it is considered of high economic importance as it causes losses at the farm level and negatively affects international trade in Europe [2]. Due to successful eradication programs, 17 of 28 European member states had gained the Official Tuberculosis Free (OTF) status by 2019 –Ireland being among the 11 non-OTF countries [3].

Ireland has been facing the challenge of eradicating bTB for a long period, and introduced a national eradication program in 1954 [4]. The first recorded data showed approximately 90,000 bTB reactors in 1959 [5] (cattle population of 4.68 million cattle; crude prevalence 1.92% [6]), a number that steadily declined to approximately 17,500 reactors by 2018 [5] (cattle population 7.34 million cattle; crude prevalence 0.24% [6]). Despite this promising reduction, bTB has not yet been fully eradicated from Ireland, which is considered to be partially attributable to the spill-over infection to and from an abundant wildlife reservoir, the European badger (*Meles meles*) [7] and maybe deer [8]. Recently, a rise in herd incidence of bTB infection from the lowest level of 3.27% in 2016 to 4.37% in 2020 has been observed [9]. This increasing incidence has triggered the Irish government to launch a renewed eradication strategy: 'Bovine TB Eradication Strategy 2021–2030' with the goal of driving down bTB levels by 2030 towards eradication [9, 10]. The program includes active surveillance with annual herd testing with the single intradermal comparative tuberculin test (SICTT), slaughterhouse (factory) surveillance at the slaughter line to detect bTB gross lesions from antemortem test negative animals, termed factory lesions (FL), as well as the removal (culling) and vaccination of badgers [11].

The annual SICTT targets all bovines over six weeks old, and combined with culling programs it aims to detect and eliminate reactor animals in the early stage of infection or pre-clinical stage [12]. Next to culling reactors, the herd of origin is restricted from trade and repeatedly tested at sixty-day intervals, in addition to herd-specific epidemiological investigations and supplementary testing where appropriate. If two consecutive herd tests are negative, herds regain an Official TB Freedom (OTF) status and trade restrictions will be lifted [10]. However, herds with OTF status still have the probability of having *M. bovis* infected animals due to the imperfect sensitivity of the SICTT [9], which has been reported as 90.6% [13] and between 52.9% to 60.8% [14] under Irish conditions using traditional and Bayesian approaches, respectively. In a meta-analysis of studies based on data from Ireland and the UK, a sensitivity of 50% (95% CrI 0.26, 0.78) and a median specificity of 100% (95% CrI 0.99, 1.00) was estimated [15]. Therefore, routine inspection by certified veterinarians for presence of bTB-like lesions in organs, head and thoracic lymph nodes of slaughtered animals is an

essential part of bTB surveillance program in Ireland. When an FL is suspected to be due to *M. bovis*, a sample is taken and submitted to the national TB laboratory for confirmation [10]. Detection of FL has been indicated as a crucial component in the surveillance program as it can disclose bTB infections in attested cattle–i.e., in cattle that originate from OTF herds which are not thought to be infected at the time they are sent to slaughter [16]. In the period 1989–1997, slaughterhouse surveillance disclosed annually 9% to 33% of all herd breakdowns in Ireland [17], while in a recent report, 20,116 bTB breakdowns in the 2005 to 2019 period were commenced by the disclosure of factory lesions [18]. In Northern-Ireland, this percentage was between 18% and 28% [19].

Slaughterhouse surveillance to detect FL at slaughter is part of the bTB eradication program in many countries, not only in Ireland but also in Great Britain [20], Northern Ireland [19, 21], Spain [22], and the United States [23]. Slaughterhouse surveillance based on meat inspection (syn: post-mortem inspection) also has been used to investigate disease epidemiology and traceback investigations of others diseases such as fascioliasis [24] and bovine cysticercosis [25]. Detection of bTB lesions at slaughter is not only detecting part of the infected cattle that remained undetected from routine skin testing but also as a tool for early detection of infected herds [19], and as a 'whistle-blower' to prevent the spread of infection to other herds. Continuous and regular monitoring of animal, herd and factory-related factors through slaughterhouse surveillance is important as quality control of the bTB eradication program [26].

As slaughterhouse surveillance is an important method for detecting bTB infections, its effectiveness needs to be assessed regularly [19]. Failure to detect infected animals at slaughter will potentially enable the further transmission of bTB infection from and within the herds the slaughtered animals originate from, especially in endemic areas with a low frequency of skin testing as is the case in parts of Great Britain [27]. Several studies [16, 19, 22, 26] have been conducted to measure the variation between factories in the proportion of slaughtered animals from which a sample was submitted to the laboratory ('submission risk') and the variation in the proportion of those samples that were subsequently confirmed to be due to *M. bovis* ('confirmation risk). For the period 2003–2004, submission and confirmation risks in Ireland were at average 0.22% and 64.4% [16], for the period 2005–2007 these were 0.25% and 68.5% [26]. The submission risk varied from 0.08% to 0.58% [16] and from 0.03% to 0.53% [26] between factories that have slaughtered at least 10,000 animals. This variation could be due to confounding factors, i.e., age, sex, the herd of origin, the season of slaughter, bTB history of the herd, and the geographical risk that were taken into account in the estimation of the factory's relative probability to submit a sample. Risk factors associated with FL detection might differ over time due to changes in the composition of the national cattle population [28] and epidemiological situation [5].

Therefore, the objective of this study was to reassess the relative effectiveness of Irish factories in submitting FL and the confirmation rate of those FLs from slaughtered cattle over the period 2014–2018. Also, risk factors at the animal, herd, and factory level for the submission and confirmation of FL were identified and quantified. The results from this study provide an evidence base for the evaluation of the effectiveness of this component of bTB surveillance in Ireland. Next to that, it will inform policy development and operational oversight to improve the detection of infected herds, support the updating of the bTB eradication programme and contribute to the reduction of bTB levels in Irish cattle.

## Materials and methods

### Dataset

The data for this study were obtained from the Animal Health Computer System (AHCS) of Department of Agriculture, Food and Marine (DAFM), Ireland. The original dataset includes

information on 6,611,854 animals slaughtered in 35 Irish factories from 2014 to 2018. The dataset contains records of animal data, tuberculin tests, and laboratory results. The recorded animal data includes animal identification numbers, age, sex, breed, date of birth (DOB), date of death (DOD), and whether a lesion was detected and submitted for laboratory confirmation. Data on the herd history of annual tuberculin testing based on the annual SICTT concerned the number of animals tested per herd, last testing date, herd size, herd type, herd identification number, and herd location based on county. The carcasses of slaughtered animals are routinely examined by veterinary inspectors for presence of lesions, as part of standard food safety post-mortem veterinary examinations. The inspection included visual examination, palpation, and incision of several organs (e.g., lungs, heart, and liver) and lymph nodes such as parotid, submandibular, retropharyngeal, apical, bronchial, tracheobronchial, mediastinal, hepatic, and mesenteric lymph nodes. When a suspected lesions was observed, tissue samples were collected and sent to the laboratory, where samples were processed for histopathological staining and/or bacterial culture to confirm bTB infection.

## Data analysis

All collected data for this study were double-checked for redundancy, missing values and then assembled for analysis. Due to many animal breeds, breed was categorized as cross breed and pure breed. Animal movement was categorized into purchased and homebred animals based on the herd identification number. If the animal was born and sent to slaughter from the birth herd, it was categorized as "homebred", if not as "purchased". All potential risk factors included in the statistical analysis were categorized as shown in Table 1.

The analytical method was in accordance with previous studies [16, 26]. The relative performance of each factory in detecting bTB lesions was based on the submission of suspected lesions and the laboratory confirmation of submitted lesions as bTB positive. In the first step, descriptive analysis was carried out to summarize the risk of FL submission and confirmation per factory, and the Spearman rank correlation between submission and confirmation risk was estimated. In addition, the frequency distribution and proportion of submitted and confirmed FL were calculated for each category of the risk factors. All data editing and descriptive analyses were performed in SAS on Demand (SAS Institute Inc). The effect of each risk factor on

**Table 1. Potential risk factors considered to have associations with FL submission and confirmation from animals slaughtered in Irish factories from 2014–2018.**

| Risk factors | Categories |
|---|---|
| **Factory** | **Per factory** |
| Year | Per year slaughter (2014–2018) |
| Month of slaughter | Per month (January–December) |
| **Animal-related factors** | |
| Age (year) | Per year category (1 to $\geq$ 8 years) |
| Sex | Male; Female |
| Animal movement | Homebred; Purchased |
| Breed | Cross breed; Pure breed |
| Ever inconclusive | Yes; No |
| Ever severe inconclusive | Yes; No |
| **Herd-related factors** | |
| Herd type | Beef; Dairy; Others; Suckler; |
| Herd size | Log transformed |
| BTB history (past 3 years) | Yes; No |

the FL submission and confirmation risk was analysed with univariable logistic regression (with herd as random effect), followed by multivariable mixed effects logistic regression using Stata SE 16.1 (Stata Corp, USA), see Eq 1.

$$LN\left(\frac{\pi_{ij}}{1 - \pi_{ij}}\right) = \beta_0 + \beta_i F_i + \beta_n C_n + \mu_j \tag{1}$$

Where $\pi_{ij}$ is the probability of having a lesion submitted (or confirmed if submitted) if an animal of herd $j$ is slaughtered in factory $i$; $\beta_0, \beta_i, \beta_n$ are the regression coefficients for the intercept, the factories (F) and all the covariates (C) respectively and where $\mu_j$ represent the random effect of herd j. In the univariable model the covariates were not included.

The crude and adjusted odds ratio (OR) of each factory, as obtained from univariable and multivariable analysis, were ranked accordingly from the highest to lowest OR, and a comparison of crude and adjusted ranking were made to determine whether the adjustment with potential confounding risk factors and herd effect affected the OR estimates and the ranking of factories. The comparison of crude and adjusted factory ranking and OR were visualized in forest plots using Microsoft Excel (Microsoft Corporation, USA).

## Results

### Descriptive results

The dataset of routine slaughterhouse surveillance consisted of 6,611,854 animals slaughtered in 35 Irish factories from 2014 to 2018 (S1 Table). Records of 33,609 (0.51%) animals were not used in the analysis: 6202 animals due to four small factories (slaughter <10,000 animals during the study period), 2716 animals due to missing values for at least one variable (i.e., age, number of skin tests, breed, herd type, herd size), and 24,901 animals were below one-year-old (some of these animals had combined reasons of exclusion with small factory and missing values). Twenty-five (0.07%) of excluded animals had FL submitted, and 16 (64%) of those FL were confirmed as bTB positive. The final dataset for statistical analysis consisted of 6,578,245 animals originating from 84,429 herds that were slaughtered in 31 Irish factories. Many herds sent their animals to more than one factory during the study period (up to 17 factories). A total of 13,337 (0.20%) animals were suspected of having FL, of which 5431 (40.72%) were confirmed as bTB positive in the laboratory. The correlation between crude submission and confirmation risk of the factories were not statistically significant (Spearman $\rho$ = -0.074, P = 0.69). The crude submission risk per factory ranged from 0.01 to 0.43%, average 0.20% and the confirmation risk of submitted lesions as bTB positive ranged from 0.00 to 61.84%, average 40.72% (Table 2). The submission and confirmation risk of factories also varied over years (Figs 1 and 2). Overall, the proportion of bTB confirmed animals based on FL submission was 0.08% in the 2014–2018 period.

The majority of animals slaughtered in Irish factories were two years old (48.50%), and 8.34% were equal to or more than eight years old. The submission risk increased as the animal gets older, ranging from 0.15 to 0.42%, while the confirmation risk ranged from 28.31 to 60.55%. Female animals had a higher submission and confirmation risk than male animals (0.25 vs. 0.17% and 48.05 vs. 31.02%, respectively). The average age at slaughter of females was 54.25 months and 27.03 months for males. Purchased animals had slightly higher submission and confirmation risk than homebred animals (0.21 vs. 0.20% and 41.33 vs. 39.21%, respectively). A total of 11,526 (0.18%) of slaughtered animals had at least one inconclusive result in routine tuberculin testing, and the submission risk of these animals was strongly increased (14.34 vs. 0.18%), although the confirmation risk was lower compared to animals that never

**Table 2. Distribution of animals slaughtered, submission risk, and confirmation risk of submitted lesions over factories included in the analysis.**

| Factory | Total slaughtered | Number submission | Submission risk (%) | Number confirmation | Confirmation risk (%) |
|---|---|---|---|---|---|
| S01 | 222,696 | 952 | 0.43 | 348 | 36.55 |
| S02 | 73,089 | 268 | 0.37 | 130 | 48.51 |
| S03 | 310,734 | 1031 | 0.33 | 492 | 47.72 |
| S04 | 210,407 | 627 | 0.30 | 273 | 43.54 |
| S05 | 258,947 | 735 | 0.28 | 239 | 32.52 |
| S06 | 232,000 | 621 | 0.27 | 148 | 23.83 |
| S07 | 28,844 | 76 | 0.26 | 24 | 31.58 |
| S08 | 296,572 | 752 | 0.25 | 300 | 39.89 |
| S09 | 319,900 | 801 | 0.25 | 366 | 45.69 |
| S10 | 316,175 | 756 | 0.24 | 392 | 51.85 |
| S11 | 386,745 | 888 | 0.23 | 301 | 33.90 |
| S12 | 238,572 | 542 | 0.23 | 257 | 47.42 |
| S13 | 224,124 | 495 | 0.22 | 189 | 38.18 |
| S14 | 233,147 | 479 | 0.21 | 244 | 50.94 |
| S15 | 225,286 | 433 | 0.19 | 103 | 23.79 |
| S16 | 69,870 | 132 | 0.19 | 48 | 36.36 |
| S17 | 229,438 | 428 | 0.19 | 115 | 26.87 |
| S18 | 250,963 | 416 | 0.17 | 146 | 35.10 |
| S19 | 16,907 | 28 | 0.17 | 9 | 32.14 |
| S20 | 333,640 | 489 | 0.15 | 154 | 31.49 |
| S21 | 229,964 | 310 | 0.13 | 162 | 52.26 |
| S22 | 307,640 | 412 | 0.13 | 240 | 58.25 |
| S23 | 106,646 | 141 | 0.13 | 61 | 43.26 |
| S24 | 51,524 | 65 | 0.13 | 19 | 29.23 |
| S25 | 281,415 | 350 | 0.12 | 137 | 39.14 |
| S26 | 262,645 | 308 | 0.12 | 131 | 42.53 |
| S27 | 145,368 | 152 | 0.10 | 94 | 61.84 |
| S28 | 223,734 | 209 | 0.09 | 76 | 36.36 |
| S29 | 366,590 | 338 | 0.09 | 175 | 51.78 |
| S30 | 111,237 | 101 | 0.09 | 58 | 57.43 |
| S31 | 13,426 | 2 | 0.01 | 0 | 0 |
| Total | 6,578,245 | 13,337 | 0.20 | 5431 | 40.72 |

had an inconclusive test result (32.49 vs. 41.89%). 17,877 (0.27%) of the slaughtered animals had been tested at least once as severe inconclusive result, and their submission and confirmation risk increased compared to animals that never had a severe inconclusive test (1.40 vs. 0.20% and 85.60 vs. 39.86%, respectively). Submission and confirmation risk were higher in animals originating from herds with bTB history in the past three years compared to animals originating from herds without bTB history in the past three years (0.41 vs. 0.12% and 69.73 vs. 2.44%, respectively). The descriptive results for all potential risk factors are presented in Table 3.

## Modeling outcomes

**Submission.** The submission OR of factories in the univariable and multivariable analysis (including all covariates) with herd as random effect are shown in Fig 3. The relative factory ranking did not change greatly: only 6 factories swapped between quartiles of the ranking. The

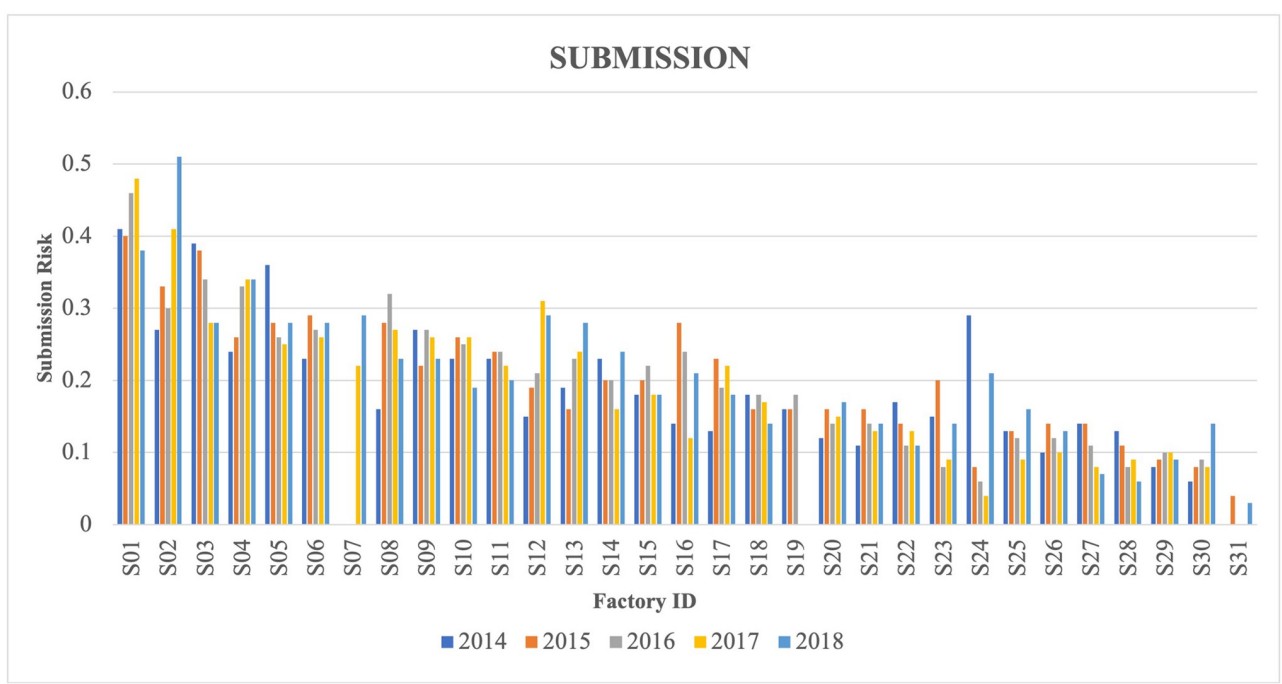

**Fig 1. Submission risk (percentage) of lesions for 31 Irish factories per year (2014–2018).**

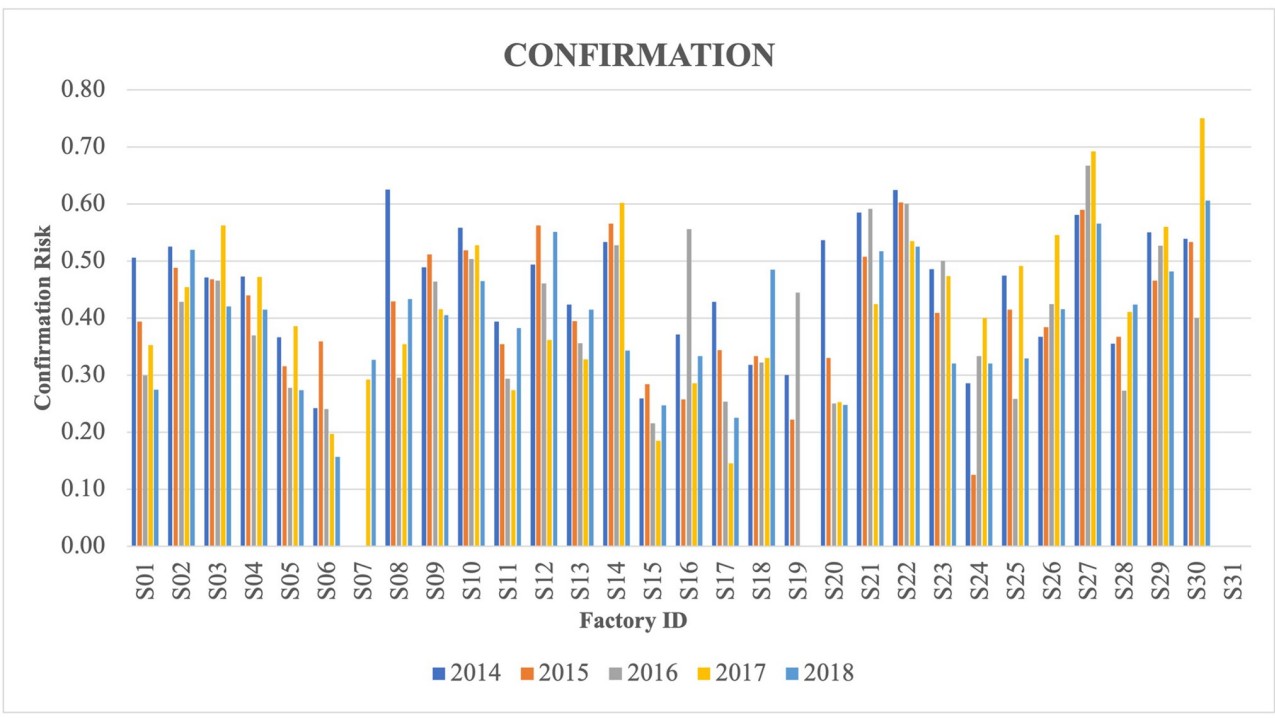

**Fig 2. Confirmation risk (proportion) of lesions submitted by 31 Irish factories per year (2014–2018).**

**Table 3. Univariable and multivariable (adjusted for covariates and random herd effect) logistic regression results of submitted (n = 6,578,245) and confirmed (n = 13,337) bTB lesions from animals slaughtered in Irish factories in the period 2014–2018.**

| Variable | Number slaughter | Submission (%) | Confirmation (%) | Univariable | | Multivariable | |
|---|---|---|---|---|---|---|---|
| | | | | Submission OR; 95% CI | Confirmation OR; 95% CI | Submission OR; 95% CI | Confirmation OR; 95% CI |
| *Year* | | | | | | | |
| 2014 | 1,354,005 | 0.20 | 46.14 | Ref. | Ref. | Ref. | Ref. |
| 2015 | 1,273,270 | 0.21 | 42.40 | 1.08; 1.02–1.14 | 0.79; 0.67–0.91 | 1.13; 1.07–1.20 | 0.81; 0.67–0.98 |
| 2016 | 1,323,348 | 0.21 | 37.85 | 1.08; 1.02–1.14 | 0.62; 0.53–0.72 | 1.16; 1.10–1.23 | 0.73; 0.60–0.88 |
| 2017 | 1,399,721 | 0.20 | 39.38 | 1.05; 1.00–1.11 | 0.70; 0.60–0.81 | 1.13; 1.07–1.19 | 0.82; 0.67–0.99 |
| 2018 | 1,227,901 | 0.20 | 37.87 | 1.05; 1.00–1.11 | 0.65; 0.56–0.76 | 1.13; 1.06–1.20 | 0.80; 0.66–0.99 |
| *Age (year)* | | | | | | | |
| 1 | 1,845,788 | 0.15 | 28.13 | Ref. | Ref. | Ref. | Ref. |
| 2 | 3,190,707 | 0.16 | 37.07 | 1.11; 1.06–1.17 | 1.68; 1.46–1.94 | 1.11; 1.05–1.17 | 1.75; 1.46–2.09 |
| 3 | 363,954 | 0.28 | 36.37 | 1.91; 1.77–2.06 | 1.64; 1.33–2.03 | 1.73; 1.59–1.87 | 1.38; 1.05–1.80 |
| 4 | 176,970 | 0.31 | 42.86 | 2.20; 2.00–2.42 | 2.39; 1.84–3.11 | 1.84; 1.66–2.04 | 2.3; 1.62–3.26 |
| 5 | 160,529 | 0.31 | 41.18 | 2.18; 1.97–2.40 | 2.22; 1.69–2.91 | 1.70; 1.53–1.90 | 1.71; 1.20–2.44 |
| 6 | 150,557 | 0.35 | 48.66 | 2.47; 2.25–2.72 | 3.22; 2.47–4.20 | 1.88; 1.69–2.09 | 3.06; 2.12–4.40 |
| 7 | 140,981 | 0.36 | 54.69 | 2.52; 2.29–2.78 | 4.54; 3.46–5.97 | 1.86; 1.67–2.08 | 3.42; 2.38–4.93 |
| ≥8 | 548,759 | 0.42 | 60.55 | 2.91; 2.75–3.09 | 6.28; 5.25–7.52 | 1.91; 1.77–2.06 | 4.05; 3.17–5.18 |
| *Sex* | | | | | | | |
| Female | 3,017,843 | 0.25 | 48.50 | Ref. | Ref. | Ref. | Ref. |
| Male | 3,560,402 | 0.17 | 31.03 | 0.67; 0.64–0.70 | 0.36; 0.32–0.40 | 1.05; 1.00–1.10 | 0.66; 0.56–0.76 |
| *Animal movement* | | | | | | | |
| Homebred | 1,777,728 | 0.21 | 39.21 | Ref. | Ref. | Ref. | Ref. |
| Purchased | 4,800,517 | 0.20 | 41.33 | 0.97; 0.93–1.01 | 1.15; 1.02–1.28 | 0.96; 0.91–1.01 | 0.96; 0.81–1.14 |
| *Breed* | | | | | | | |
| Cross breed | 4,716,660 | 0.20 | 39.82 | Ref. | Ref. | Ref. | Ref. |
| Pure breed | 1,861,585 | 0.21 | 42.96 | 1.03; 0.99–1.08 | 1.22; 1.10–1.37 | 0.80; 0.76–0.83 | 1.04; 0.89–1.22 |
| *Ever inconclusive* | | | | | | | |
| No | 6,566,719 | 0.18 | 41.89 | Ref. | Ref. | Ref. | Ref. |
| Yes | 11,526 | 14.34 | 32.49 | 99.04; 93.19–105.26 | 0.64; 0.55–0.75 | 74.24; 69.39–79.43 | 0.44; 0.36–0.54 |
| *Ever severe inconclusive* | | | | | | | |
| No | 6,560,368 | 0.20 | 39.86 | Ref. | Ref. | Ref. | Ref. |
| Yes | 17,877 | 2.17 | 85.60 | 6.89; 6.02–7.81 | 17.94; 11.27–28.54 | 2.83; 2.44–3.27 | 4.48; 2.66–7.54 |
| *Herd type* | | | | | | | |
| Beef | 2,435,812 | 0.20 | 40.74 | Ref. | Ref. | Ref. | Ref. |
| Dairy | 1,267,432 | 0.20 | 37.91 | 1.06; 1.00–1.12 | 0.88; 0.75–1.02 | 0.83; 0.77–0.90 | 1.08; 0.85–1.38 |
| Other | 520,788 | 0.18 | 33.19 | 0.85; 0.78–0.93 | 0.61; 0.49–0.77 | 0.99; 0.90–1.08 | 1.11; 0.81–1.51 |
| Suckler | 2,354,213 | 0.21 | 43.51 | 1.13; 1.07–1.18 | 1.21; 1.07–1.38 | 1.08; 1.02–1.14 | 1.31; 1.11–1.55 |
| *Herd size* | | | | 0.98; 0.96–0.99 | 0.96; 0.59–0.81 | 0.95; 0.93–0.96 | 0.82; 0.78–0.86 |
| *BTB history (past 3 years)* | | | | | | | |
| No | 4,722,722 | 0.12 | 2.44 | Ref. | Ref. | Ref. | Ref. |
| Yes | 1,855,523 | 0.41 | 69.73 | 4.87; 4.66–5.09 | 355.80; 2.65–478.26 | 4.46; 4.28–4.66 | 319.90; 237.98–430.04 |
| *Months of slaughter* | | | | | | | |
| January | 562,269 | 0.19 | 37.55 | Ref. | Ref. | Ref. | Ref. |
| February | 430,767 | 0.18 | 42.56 | 0.93; 0.85–1.02 | 1.31; 1.01–1.71 | 0.94; 0.86–1.04 | 1.10; 0.79–1.53 |
| March | 562,055 | 0.18 | 39.36 | 0.97; 0.89–1.06 | 1.09; 0.85–1.38 | 0.98; 0.90–1.07 | 1.13; 0.83–1.53 |

*(Continued)*

**Table 3.** (Continued)

| Variable | Number slaughter | Submission (%) | Confirmation (%) | Univariable | | Multivariable | |
|---|---|---|---|---|---|---|---|
| | | | | Submission OR; 95% CI | Confirmation OR; 95% CI | Submission OR; 95% CI | Confirmation OR; 95% CI |
| April | 527,150 | 0.20 | 38.06 | 1.07; 0.98–1.16 | 1.06; 0.83–1.34 | 1.04; 0.95–1.14 | 1.01; 0.74–1.36 |
| May | 534,771 | 0.19 | 38.73 | 1.02; 0.93–1.11 | 1.15; 0.90–1.46 | 0.94; 0.86–1.03 | 1.02; 0.75–1.39 |
| June | 536,051 | 0.21 | 40.37 | 1.11; 1.02–1.21 | 1.07; 0.84–1.36 | 1.01; 0.93–1.10 | 0.84; 0.62–1.13 |
| July | 558,296 | 0.22 | 40.87 | 1.16; 1.07–1.26 | 1.24; 0.98–1.56 | 1.05; 0.96–1.15 | 1.11; 0.83–1.50 |
| August | 588,359 | 0.21 | 39.34 | 1.09; 1.01–1.19 | 1.12; 0.89–1.42 | 1.02; 0.94–1.11 | 1.01; 0.75–1.35 |
| September | 622,015 | 0.21 | 41.42 | 1.10; 1.01–1.19 | 1.23; 0.98–1.55 | 1.05; 0.97–1.15 | 1.14; 0.85–1.53 |
| October | 617,809 | 0.20 | 42.92 | 1.05; 0.97–1.14 | 1.29; 1.02–1.63 | 1.01; 0.93–1.10 | 1.39; 1.03–1.87 |
| November | 626,507 | 0.20 | 42.93 | 1.04; 0.95–1.13 | 1.38; 1.10–1.75 | 0.99; 0.91–1.08 | 1.34; 1.00–1.81 |
| December | 412,196 | 0.22 | 44.86 | 1.14; 1.04–1.25 | 1.48; 1.16–1.91 | 1.11; 1.02–1.22 | 1.55; 1.13–2.13 |
| Total | 6,578,245 | 0.20 | 40.72 | | | | |

maximum change in factory ranking was 6 positions and the average being 2 positions (S2 Table). Herd as random effect explained 15% of the residual variance (Intraclass correlation (ICC) = 0.15; 95% CI 0.14–0.16) in the multivariable model. The odds of FL submission increased with the increase of animal's age at slaughter, and the OR of animals over eight years old was 1.91 compared to animals of 1–2 years old. Males had lower odds of having FL submitted than females (OR 0.67, 95% CI 0.64–0.70) in the univariable analysis, but the odds increased to 1.05 in the multivariable analysis. Ever inconclusive tested animals under standard interpretation had 74.24 (95% CI 69.39–79.43) times higher odds of FL submission compared to never inconclusive animals in the multivariable analysis. The odds were much lower for animals that were defined as inconclusive based on the severe interpretation of the tuberculin skin test (OR: 2.83; 95% CI 2.44–3.27). Pure breed animals had lower odds of FL submission (OR 0.80; 95% CI 0.76–0.83) than cross breed animals. Animals originating from suckler herds had slightly higher odds of having submitted FL than animals from beef herds (OR 1.08, 95% CI 1.02–1.14), while dairy herds had lower odds of having FL (OR: 0.83; 95% CI 0.77–0.90). Herd size was negatively associated with FL submission; an increase in log herd size by one unit (mean 4.3, median 4.6, maximum 7.6) reduced the odds with a factor 0.95 (95% CI 0.93–0.96). A history of bTB breakdown in the past three years increased the odds of having a submitted FL (OR 4.46; 95% CI 4.28–4.66). The submission OR for all potential risk factors are presented in Table 3.

**Confirmation.** The confirmation ORs of factories of the univariable and multivariable model (including all covariates) are shown in Fig 4. The relative ranking of some factories substantially changed after adjustments for covariates: nine factories swapped between quartiles of the ranking and the change in factory ranking was up to 26 positions, the average being 4 positions (S2 Table). 30% of the residual variance of the multivariable model was explained by the herd effect (Intraclass correlation (ICC) = 0.30; 95% CI 0.26–0.35). Submitted FL from older animals had higher odds of being confirmed as bTB positive in the laboratory. The odds of confirming FL in animals over eight years old were 4.05 times (95% CI 3.17–5.18) higher than the odds of animals 1–2 years old. Males had lower odds of having FL confirmed than females (OR 0.66; 95% CI 0.56–0.76). The confirmation odds were also lower for FL originating from animals with inconclusive results of tuberculin skin tests (OR 0.44; 95% CI 0.36–0.54), although the odds of confirming FL from severe inconclusive animals was increased with a factor 4.48 (95% CI 2.66–7.54). The confirmation odds were slightly higher for FL originated

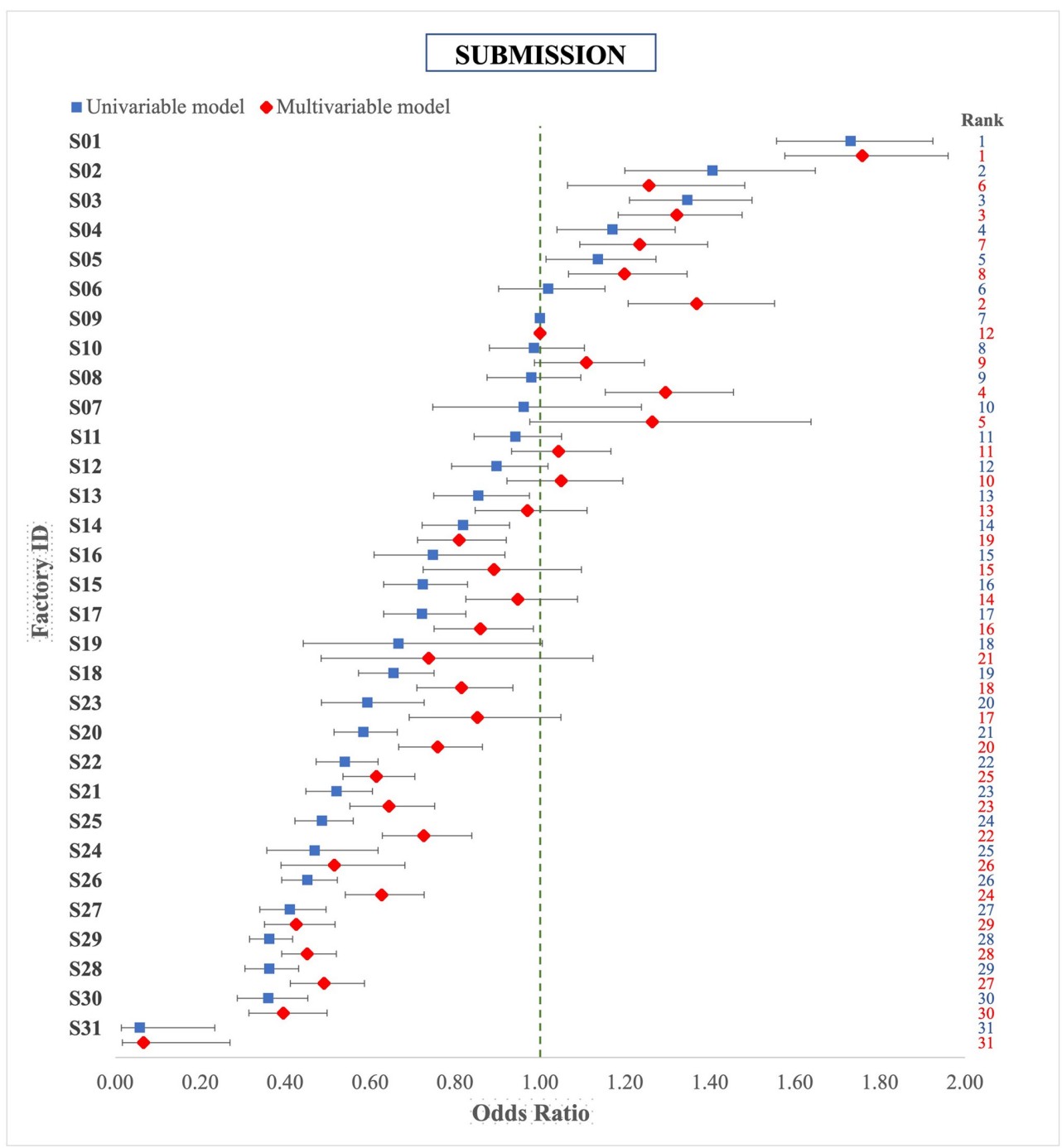

**Fig 3. Comparison of factory ranking based on OR derived from univariable and multivariable (adjusted for covariates and random herd effect) logistic regression results of bTB suspected lesion submission from animals slaughtered (n = 6,578,245) in Irish factories in the period 2014–2018.** S09 = reference category.

from cross breed animals (OR 1.04; 95% CI 0.89–1.22) than pure breed, although it was not statistically significant. FL from animals of suckler and dairy herds had 1.31 (95% CI 1.11–1.55) and 1.08 (95% CI 0.85–1.38) higher odds to be confirmed as bTB positive than FL from animals of beef herds. Increasing log herd size by one unit (mean 4.3, median 4.6, maximum

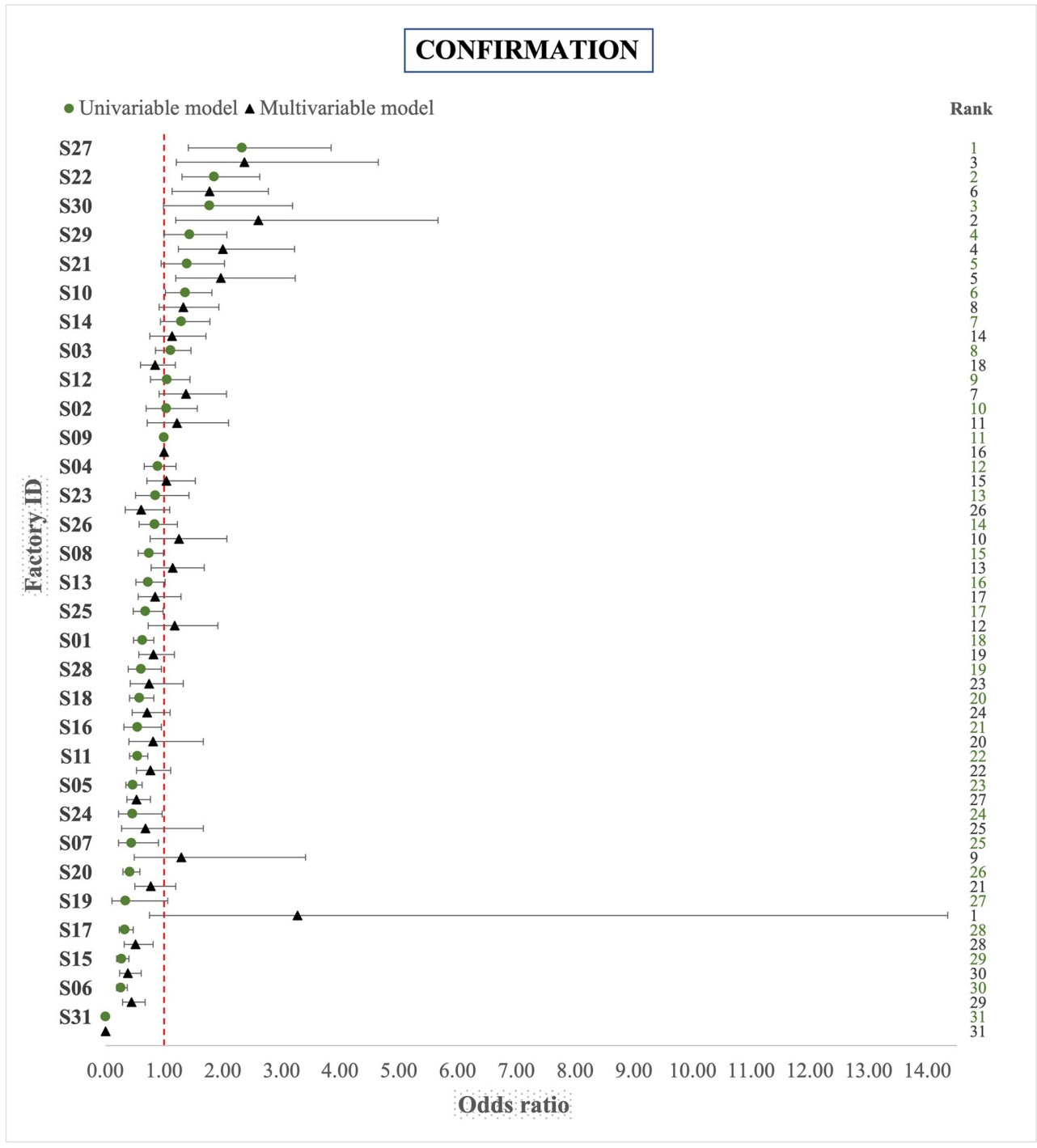

**Fig 4. Comparison of factory ranking based on OR derived from univariable and multivariable (adjusted for covariates and random herd effect) logistic regression results of bTB confirmation of lesions submitted from animals slaughtered (n = 13,337) in Irish factories in the period 2014–2018.** S09 = reference category.

7.6) lowered the odds of FL confirmation by a factor 0.82 (95% CI 0.78–0.86). Compared to the herds with no bTB breakdown in the past three years, the odds of confirming an FL were 319 times (95% CI 237.98–430.04) higher in animals from herds with recent bTB history. The confirmation OR for all potential risk factors are presented in Table 3.

## Discussion

The main aim of this study was to quantify differences in the relative effectiveness of Irish factories in detecting suspected bTB lesions among the animals slaughtered from 2014 to 2018. This study also assessed the association between potential risk factors and the submission and confirmation risk of submitted lesions.

The submission risk of FL in 31 Irish factories ranged from 0.01 to 0.43%, which means that out of 10,000 slaughtered animals, 1 to 43 animals that had a suspected FL were submitted to the laboratory for bTB confirmation, the average being 20 animals per 10,000 slaughtered animals. Similar results were obtained in a previous study over the period 2003–2004 from which 0 to 58 animals with an average of 22 animals per 10,000 slaughtered animals was reported [16]. In the period 2005 to 2007, these numbers were 0 to 52 animals with an average of 25 animals per 10,000 slaughtered animals [26]. A nine-fold, seven-fold, and five-fold increase after exclusion of factories that submitted less than 10 animals were reported from earlier studies [16, 26, 29]. This is in line with the current study as the difference is around 5-fold (0.09 vs. 0.42%) if the same criteria are used (Table 2). Furthermore, the confirmation risk of submitted FL ranged from 0.00 to 61.84% with an average of 40.72%, which is lower than the average confirmation risk reported in those previous studies of 67.2% [30], 63% [16], and 67.5% [26]. The relatively lower confirmation risk in the present study is consistent with lower bTB levels in Ireland during the study period compared to those previous study periods.

The FL submission and confirmation risk varied between factories in the univariable and multivariable analysis, and might show unequal practices in lesions detection between factories [26]. The crude and adjusted submission ranking of factories were not much different, which indicates that herd and animal factors did not contribute substantially to the variation of submission risk. However, a substantial difference in crude and adjusted confirmation ranking of some factories may indicate that animal and herd-related factors contribute substantially to the variation in FL confirmation between factories. From previous studies [16, 26, 30] it was concluded that the variation of submission and confirmation risk was not related to animal and herd-related factors, but to assigned factory-related factors that may influence the inspection performances in the slaughterhouse. Such factors include line speed, lighting, equipment, and competency of the veterinary inspectors [31, 32]. Competencies and skills of the veterinary inspectors in each factory in recognizing potential bTB lesions are important during the meat inspection. Suboptimal performance by inspectors might lead to missed lesions and the herds with truly infected animals remaining undetected. This especially may occur if the infection is in the initial stage where granulomatous lesions are frequently absent or relatively small, making it hard to detect lesions by the naked eye [33].

Several animal and herd-related factors have shown to influence bTB infection [16,19, 20, 26, 34], we are cognisant that these factors (e.g., herd type, breed, sex, and age) are somewhat intertwined. In this study, as expected, FL are more likely to be detected and confirmed in older animals, which in agreement with the findings of [16, 26, 30] in the Irish setting. Older animals will have spent a longer time in the herd, which prolonged exposure to infected animals and contaminated environment [16, 34, 35]. In addition, because bTB is a chronic disease, a longer lifespan gives more time to develop visible granulomatous lesions [30]. An inverse relationship of FL submission and confirmation risk was present in male and female

animals, and this is potentially confounded by age. FL were more likely to be disclosed in males; however, the confirmation odds were significantly lower. Females have been reported to have higher odds of being bTB positive, whether detected by routine skin testing [36] or at slaughter [16]. Different submission and confirmation risks between sex are possibly due to different management practices; for instance, dairy cows usually have a longer lifespan due to milk production [37] and therefore may have more developed lesions which are more easily detected or confirmed at slaughter.

Animals that ever had at least once inconclusive or severe inconclusive tuberculin skin result had higher odds to have a submitted FL. FL from ever inconclusive animals had lower odds to be confirmed as bTB positive (OR = 0.44); in contrast, the confirmation odds was higher for FL from severe inconclusive animals (OR = 4.48) (Table 3). A possible explanation is the policy in handling inconclusive animals. In Ireland, during the study period, animals that had an inconclusive skin result under standard interpretation must be either re-tested 42 days later or slaughtered. If farmers choose the option to have them slaughtered, lymph nodes are submitted to the laboratory regardless of the presence of visible lesions, which explains the increased submission risk and the lower confirmation risk (pers. comm. Philip Breslin, DAFM).

In the models we fitted a breed variable representing whether animals were reported within the AHCS dataset as a "pure breed" or "cross breed". Ideally, the actual breed or breed functional class (e.g. dairy breed, beef breed, dual purpose) could have been used, but exploratory analysis suggested that there was strong corelations between these classifications and "herd type" variable. Therefore, to avoid variation inflation, we instead dichotomised breed based on their reported genetics mix. It should be noted that this variable may have some limitations given the data on breed heritage which is self-reported by the farmer within the AHCS dataset. Furthermore, it is not fully known whether breed crosses are more or less susceptible to *M. bovis* infection, or that they differ in disease progression. However, some evidence suggests differences in breed heterozygosity were associated with lower susceptibility to bTB in Ireland [38], and in prevalence between crosses and local breeds in Ethiopia [39] and India [40], for example.

The odds of FL submission and confirmation were negatively associated with herd size. A similar conclusion was made in previous research undertaken at animal-level [35, 41, 42]. This is in contrast with the well-known increased risk of herd-level bTB breakdown in larger herds [37, 43]. Higher odds of having FL in smaller herds is suggested to be related to management practices in Ireland, where animals in small herds usually have close contact with other animals on the farm [41], while the larger herds are being split in cohorts and reduce the contact with infected animals [35]. Animals from suckler and dairy herds were more likely to have FL confirmed as bTB positive than beef herds. This can be due to the genetics or cattle breed [44], longer contact of suckler calves and their dam [35], or due to the longer lifespan of dairy cattle [37]. The history of bTB infection was significantly related to the submission and confirmation risks of FL. Animals from herds with a history of bTB in the past three years were more likely to have a submitted and confirmed FL. A comparable finding was obtained by [16]; however, that study considered the history of OTF years where animals from herds with 0–3 years OTF history have higher odds of having lesions confirmed than animals from herds >4 years OTF history. Herds with a history of bTB also had a higher risk for restriction in the future [45]. In the herds with bTB history, it may be possible that infected animals remain undetected with the routine skin tests and transmit the infection to other animals before they are slaughtered. It might also be due to a re-introduction of the infection by the purchase of latently infected cattle or by a wildlife reservoir.

## Conclusion

In comparison to the similar studies in Ireland during 2003–2005 [16] and 2005–2007 [26], the FL submission remained steady, but the confirmation risk was lower, consistent with a reduction in overall bTB levels in Ireland across the study periods. The relative submission ranking of factories did not change much after adjustment for animal and herd-related factors, suggesting FL submission variation between factories is mainly due to factory-related factors. The crude and adjusted ranking of some factories was substantially different for FL confirmation, which indicates that animal and herd-related factors may contribute to the variation of FL confirmation risk between factories.

## Supporting information

**S1 File.**
(ZIP)

## Acknowledgments

The authors thank the Animal Health Computer System (AHCS) database managers of the Department of Agriculture, Food and Marine (DAFM), Ireland, who provided the data for this study.

## Author Contributions

**Conceptualization:** Eoin Ryan, Philip Breslin, Andrew William Byrne.

**Formal analysis:** Rischi Robinson Male Here, Klaas Frankena.

**Methodology:** Rischi Robinson Male Here, Klaas Frankena.

**Supervision:** Klaas Frankena, Andrew William Byrne.

**Writing – original draft:** Rischi Robinson Male Here.

**Writing – review & editing:** Rischi Robinson Male Here, Eoin Ryan, Philip Breslin, Klaas Frankena, Andrew William Byrne.

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
