## [Decision Letter · Decision Letter 0]

28 Jun 2022

PONE-D-22-10323Revisiting the relative effectiveness of slaughterhouses in Ireland to detect tuberculosis lesions in cattle (2014-2018)PLOS ONE

Dear Dr. Male Here,

Both reviewers have made suggestions for improvements to your manuscript. I would pay particular attention to considering spatial aspects of your data.

We look forward to receiving your revised manuscript.

Kind regards,

Rebecca Lee Smith, D.V.M., M.S., Ph.D.

Academic Editor

PLOS ONE

Journal Requirements:

“Further thank goes to the Indonesia Endowment Fund for Education (LPDP) that granted a scholarship for RRMH to study at Wageningen University & Research”

“This study was funded by Indonesia Endowment Fund for Education (LPDP, https://lpdp.kemenkeu.go.id), Ministry of Finance, the Republic of Indonesia, as part of the scholarship awarded to RRMH. The funder had no role in study design, data collection, and analysis, decision to publish, or preparation of the manuscript.”

Reviewers' comments:

Reviewer's Responses to Questions

**Comments to the Author**

1. Is the manuscript technically sound, and do the data support the conclusions?

Reviewer #1: No

Reviewer #2: Yes

2. Has the statistical analysis been performed appropriately and rigorously? 

Reviewer #1: No

Reviewer #2: Yes

3. Have the authors made all data underlying the findings in their manuscript fully available?

Reviewer #1: No

Reviewer #2: Yes

4. Is the manuscript presented in an intelligible fashion and written in standard English?

Reviewer #1: Yes

Reviewer #2: Yes

5. Review Comments to the Author

Reviewer #1: Main considerations

The submitted manuscript reports a study on the submission rate of potentially bovine TB infected lesions from cattle slaughterhouses in Ireland, and on the subsequential bTB confirmation rates of these samples. The study aims to compare the submission and confirmation rates (they are called “risks” in the manuscript, see last point of the main considerations) across different slaughterhouses, and evaluate the risk factor which can have an influence on these rates. In particular, they compare the odds ratio of each slaughterhouse in an univariable model and with a multivariable model, including the number potential factors that might play a role in determining the slaughterhouses confirmation rates. While the subject is of interest there are some pitfalls which undermine the robustness and the depth of these findings.

First, I think there is a confusion between the concepts of “risk” and “rate”. As it stands, I found it very odd to consider that the chance of a sample to be submitted to the lab as a “risk” (same for “confirmation”). I’d suggest to change these to “submission rate” and “confirmation rate”.

A crucial but neglected aspect is the sampled population spatial distribution, i.e. where the slaughterhouses get the cattle from. Since bTB is not homogeneously spread in Ireland, the location where becomes very important. This is exacerbated by the wildlife interface spread, which makes some areas widely different to others in term of potential bTB prevalence. By neglecting so, it corresponds to assume that all the cattle are homogeneously mixed in Ireland, and that bTB is homogeneous as well, or, that all the slaughterhouses get the cattle form the same exact source. Including cattle geographical factors (e.g. farm coordinates, local TB rates) would be important for the validity of this study, otherwise comparing the slaughtrhouses submission and confirmation rates does not provide much information.

Similarly, no analysis was performed ex-post about what might cause different rates of submission and confirmation in the slaughterhouses. Do ones with similar rates cluster in space, for the type of source herds or else? It would be also interesting comparing the location of the slaughterhouses in relation to the local bTB prevalence, but I understand that location information might be subject to privacy restrictions. Nothing prevent them to be analysed, even if they are not shown on a map.

Starting by saying that I am not a statistician, I don’t understand the logic behind the model usage here. In my understanding univariable models are run with a single explanatory variable at a time, and then the most significant ones are run in a multivariable model. At first my best guess was that the univariable model included only the slaughterhouse (as a fixed effect), and that the multivariable model included all the variable in table 1 as fixed effects, and the slaughterhouses as a random effect. However, I realised that the “univariable” model included many variables indeed (see Table S4 and S5). Was the univariable model just a multivariable model but run independently for each slaughterhouse? Or was the slaughtrhouses factor just neglected in the "univariable" model? If this is the case, was the comparison between “univariable” and “multivariable” just a methodological aspect?

The difficulty in understanding what was done is also caused by the absence of any model formula. Even if these are well known methods, reporting a rigorous mathematical formula would help the readers (and the reviewers) in understanding what was done. Finally, no model metric was reported (AIC, BIC). Were all models performing in a similar way? How well these models explained the observations?

Finally, I suggest a thorough read of the manuscript before potential resubmissions.

Minor comments

General: the term “factory lesions” sounds odd, it seems like the lesions happened at the factory. I can live with that, but maybe “internal lesions” or something else could be more straightforward for the readers.

Abstract: it would be more beneficial for the readers to have some more background and broader conclusions and considerations in this section, rather than delving into the details of the results.

Line 49: in Europe only?

Line 56: remove “thus”.

Line 58: I know that the presence of bTB deer is still controversial, but it is worth mentioning (see Crispell et al., 2020).

Line 65: the badger culling program?

Line 81: why an interval is reported here? Is it the percentage of breakdowns per year?

Lines 80-83: would it be possible to report all three data in the same format, like all number or percentages (better) of breakdowns disclosed? Otherwise they are difficult to compare.

Line 92: remove “the” in front of Ireland.

Lines 93-94: confusing, please just report the range.

Lines 99-105: these sentences are convoluted and hard to read.

Line 113: is DOD because of “date of death”?

Line 127: slaughtered in the same herd or sent to slaughter from the birth herd?

Line 137: which distribution was calculated? And where is it reported?

Table 1: are slaughterhouses Z, ZA, ZB, ZC, ZD, ZE related or there are just not enough letters? Did you consider to switch to number or a code instead (S01, S01 or any)?

Table 1 and Figure 1: are the slaughterhouses in random order? Would it be more valuable for the readers if they were ranked according to number of slaughtered animals (or some other metric), instead?

Table 1: how solid is the inclusion of slaughterhouse T in the study, given only 2 samples submitted to the lab?

Lines 173-174: confusing, please rephrase.

Line 180: “these animals” are the ones with at least one inconclusive test?

Line 185: were “higher”?

Line 186: how is the 3-year threshold being selected? Does it make a difference if the 2 or 4 years would be selected to defined the past bTB history? This sensitivity analysis would be material for supplementary information.

Line 224: “lower”?

Lines 253-261: I struggle to understand the point of this paragraph, in particular 257-261. Why the VOI procedure should make any difference, if it’s not in place in Ireland?

Lines 271-275: were these factors addressed in this analysis?

Lines 284-287: herd type, breed, sex and age are intimately intertwined factors in modern cattle industry. Since they are mostly treated as stand-alone factors, I would suggest to integrate them at least in the discussion, so to provide the readers a clearer picture.

Lines 323-332: this paragraph is a repetition from the introduction, rather than discussion material.

S1: I might have missed this in the text, but why some factories have been excluded from the study?

Reviewer #2: The manuscript “revisiting the relative effectiveness of slaughterhouses in Ireland to detect tuberculosis lesions in cattle (2014-2018)” updates the analyses carried out in the past regarding the differences between slaughterhouses in Ireland in the rate of submission and laboratory confirmation of tuberculosis-compatible lesions. Given the importance of passive surveillance in the eradication programs for bovine tuberculosis in countries where the disease is at (more or less) reduced levels, the continued evaluation of its performance is an objective worth pursuing, and therefore the topic under study here has scientific merit. The analytical framework used is well established and was used in the past with the same objective, and the interpretation of the results is sound. Still, the addition of some details in terms of methodology and results could help to better understand what was performed, and what the results were.

Comments:

- Line 21-22: the term “ever inconclusive or severe inconclusive skin test reactor” is not very straightforward, suggest replacing it by “ever reactor in the standard or severe skin test” or similar (more similar to what was used in lines 30-31)

- Line 46-47: I would suggest the authors to refer to bovine tuberculosis as the disease caused by members of the M. tuberculosis complex in bovine rather than the disease caused by M. bovis in any host species (though both definitions are widely used in the literature, I think we would not refer to bovine brucellosis as the infection by B. abortus in sheep, and the former definition is more aligned with current regulations in the EU).

- Line 51: suggest replacing “in” by “by” (since countries did not gain the OTF status precisely in 2019)

- Lines 94-97: Given that in the papers mentioned in the previous lines (refs 15 and 20) at least some of the covariables mentioned here (age, gender…) were considered in the multivariable models, I would expect that variations in their distributions depending on the slaughterhouse should not explain the difference in submission risk per slaughterhouse (that is the point of considering them in the analysis, isn’t it?).

- Line 113: I think gender is a trait only applicable to humans, and sex should be used here (as e.g. used in Table 1).

- Line 114: for purchased animals: did the herd history include all the annual tuberculin testing in all herds where the animal had stayed or only the last one prior to be shipped to the abattoir? (the last testing date is mentioned in line 115 but I wonder if this refers to the whole information on skin testing).

- Data analysis: Herd size is an interesting covariable but I wonder if it makes sense to assume a linear relationship between the (logit of) the probability of submission/confirmation and the number of animals in the herd. Did the authors consider categorizing in some way this variable and see whether it improved (or not) model fit? Although categorizing a continuous variables has its own drawbacks, in terms of interpretation it may be easier to see increase in the risk in “large” vs. “small” herds (just a suggestion). Also, it seems (based on Table S4) it was log-transformed; if so, this should be indicated in the methodology section.

- Data analysis (II): I would be interested in knowing the view of the authors regarding considering factory as a random rather than as a fixed effect. Though I see the point in considering it a fixed effect, given the very large number of categories and the potential objective of characterizing variability across slaughterhouses assuming the belong to some sort of population, coupled with the view of the database as a hierarchical structure (with animals clustered in abattoirs). Note that I don’t think there is anything wrong with the approach used and I am not requesting the authors to change it, but I would like to know if this was considered at some point or if the authors have a strong view against this option.

- Line 155: I was surprised to see animals <1 year were excluded but looking at S2 I realize they represent a very small fraction of the population. In other countries it may be not so unusual to slaughter animals below one year (10-11 months-old), so perhaps the fact that animals below 1 year were excluded could be added in the material and methods as an exclusion criterion

- Line 153: suggest to specify that there are four small factories (I think there are four based on line 158, 31 factories after exclusion of small ones).

- Lines 161-162: what does this correlation refer to? The crude submission and confirmation risks per factory? This could be perhaps added as a supplementary graph (and although I assume it is considered included in the “descriptive analysis” part on the material and methods, also be specified there as an analysis performed).

- Line 175: suggest replacing “ranged” by “ranging”

- Modeling outcomes: I would suggest to start both submission and confirmation sections by indicating which variables (all considered, I pressume?) were included in the final model. Also, I would strongly suggest to add a table in the main text with the adjusted OR coefficients (along with some of the descriptives included in Table S2?) for these covariables (something similar to Table 1 in the paper by Frankena et al). Though this is included in table S4, I think it is very useful information that readers would appreciate having in the main text (and could be easily integrated in Table 1, that currently conveys only limited information).

- Line 191-192 and 216-217: given that factory-specific ORs are calculated based on the difference with the baseline abattoir, I am not sure how to interpret these sentences (differences between factories become larger once covariates were considered? But this would be only considering the reference category, wouldn’t it? Perhaps other differences decrease?)

- Line 251: suggest replacing “difference” by “increase” (since I think this is what is meant).

- Line 257-261: did the meat inspection procedure change in Ireland at some point though?

- Lines 276-287: discussion of the observed effects of age and sex is interesting. Did the authors consider a potential interaction between these covariates?

- Lines 293-295: are these lymph nodes from inconclusive cattle counted as “submitted” in the database? I guess this is an important source of error since for the rest submission will always be linked to the presence of lesions while it wouldn’t be the case for the inconclusive cattle. I would suggest to declare this more openly as a limitation of the study that – hopefully – affects a minor proportion of animals in the database (less than 20k animals in the whole database). Would these animals be preferentially submitted to certain slaughterhouse?

- Lines 302-305: this is a very long sentence that doesn’t read very easily, suggest to break it in two sentences and revise it.

- Lines 323-332: this seems more an introductory section, I would suggest to remove it (or relocate it in the introduction).

- Are (most of) the factories the same as in the previous papers by Frankena et al and Olea-Popelka et al.? If so, and is possible for the authors, it would be interesting to comment on whether factories remain in the same ranking (or quartile) over time.

- A minor suggestion is to remove the second decimal in numbers above 10% (e.g., replace 40.72% by 40.7%, since with large numbers it seems unnecessary). Also, I think sentences should not start with a number (e.g., line 159, start with e.g. “A total of 13,337” instead of by “13,337” directly after the full stop).

6. PLOS authors have the option to publish the peer review history of their article (what does this mean?). If published, this will include your full peer review and any attached files.

Reviewer #1: No

Reviewer #2: No

---

## [Author Response · Author response to Decision Letter 0]

15 Jul 2022

We appreciate very much the time and effort the reviewers have taken to read and comment on our manuscript. The comments and suggestion were very helpful to improve the paper and below you can find all our responses with indications to where we have changed the manuscript. We hope that our answers to the queries are satisfactory and that the revised version is now acceptable for publication.

On behalf of all authors,

Rischi Robinson Male Here

(corresponding author)

Reviewer #1: Main considerations

The submitted manuscript reports a study on the submission rate of potentially bovine TB infected lesions from cattle slaughterhouses in Ireland, and on the subsequential bTB confirmation rates of these samples. The study aims to compare the submission and confirmation rates (they are called “risks” in the manuscript, see last point of the main considerations) across different slaughterhouses, and evaluate the risk factor which can have an influence on these rates. In particular, they compare the odds ratio of each slaughterhouse in an univariable model and with a multivariable model, including the number potential factors that might play a role in determining the slaughterhouses confirmation rates. While the subject is of interest there are some pitfalls which undermine the robustness and the depth of these findings.

First, I think there is a confusion between the concepts of “risk” and “rate”. As it stands, I found it very odd to consider that the chance of a sample to be submitted to the lab as a “risk” (same for “confirmation”). I’d suggest to change these to “submission rate” and “confirmation rate”.

AU: by definition a rate has a dimension, e.g. per time unit, which clearly is not the case here. Here, risk is meaning ‘probability’. The description of submission and confirmation risk are stated in lines 101 and 103 as the proportion of slaughtered cows from which a sample has been submitted and the proportion of submitted samples that were confirmed. The same terminology has been used in our previous papers [1,2] and also in others [3] and we would like to stick to this, also for consistency reasons. Perhaps the reviewer is confused by the presentation as ‘per 10,000 animals’ (line 270 - 271) which is just a matter of scaling.

A crucial but neglected aspect is the sampled population spatial distribution, i.e. where the slaughterhouses get the cattle from. Since bTB is not homogeneously spread in Ireland, the location where becomes very important. This is exacerbated by the wildlife interface spread, which makes some areas widely different to others in term of potential bTB prevalence. By neglecting so, it corresponds to assume that all the cattle are homogeneously mixed in Ireland, and that bTB is homogeneous as well, or, that all the slaughterhouses get the cattle form the same exact source. Including cattle geographical factors (e.g. farm coordinates, local TB rates) would be important for the validity of this study, otherwise comparing the slaughterhouses submission and confirmation rates does not provide much information.

AU: this is very valid comment. For privacy, and in compliance with GDPR a data sharing agreement between the data owners (DAFM) and data processors (WUR) for this particular project, factories and herds were anonymised. Therefore, we only had access to the county the cow was send from to the slaughterhouse. However, 73% of the slaughtered animals have moved between herds as the herd of birth is not the same as the herd it was culled from (see table 3, animal movement). Next to that 25 of 31 factories have slaughtered animals from at least 20 (of 26) counties. So it is rather impossible to further elaborate on the spatial distribution as animals are not slaughtered as locally as one would expect and most animals have been in at least 2 herds. Moreover, the herd’s BTB history has been taken into account as covariate in the model and will represent a large part of the spatial heterogeneity, as recrudescence is a characteristic of herd-level bTB epidemiology in Ireland.

Lastly, we remind the reviewer that we are modelling animals with factory lesions, these are animals that did not fail an antemortem test but were found to have evidence of infection post-mortem. Therefore, we expect that the local area prevalence in cattle or wildlife may not have as a significant effect on the sensitivity of the test that within-herd dynamics (see e.g. [4]).

Similarly, no analysis was performed ex-post about what might cause different rates of submission and confirmation in the slaughterhouses. Do ones with similar rates cluster in space, for the type of source herds or else? It would be also interesting comparing the location of the slaughterhouses in relation to the local bTB prevalence, but I understand that location information might be subject to privacy restrictions. Nothing prevent them to be analysed, even if they are not shown on a map.

AU: Indeed variation between factories is interesting, but the reviewer is correct: the slaughterhouses were anonymised and we do not know their location. See also above where it is explained that many animals are not slaughtered locally as slaughterhouses receive animals originating from many counties.

Starting by saying that I am not a statistician, I don’t understand the logic behind the model usage here. In my understanding univariable models are run with a single explanatory variable at a time, and then the most significant ones are run in a multivariable model. At first my best guess was that the univariable model included only the slaughterhouse (as a fixed effect), and that the multivariable model included all the variable in table 1 as fixed effects, and the slaughterhouses as a random effect. However, I realised that the “univariable” model included many variables indeed (see Table S4 and S5). Was the univariable model just a multivariable model but run independently for each slaughterhouse? Or was the slaughtrhouses factor just neglected in the "univariable" model? If this is the case, was the comparison between “univariable” and “multivariable” just a methodological aspect?

AU: the primary factor of interest in this study is the factory and in the univariable model only Factory_id was included as fixed effect (i.e. all factories) and herd_id as random effect. In the multivariable model all potential covariates were added to the univariable model to see if factory rankings would change, while controlling for these herd-level characteristics of the pool of herds contributing to each factory. 

The difficulty in understanding what was done is also caused by the absence of any model formula. Even if these are well known methods, reporting a rigorous mathematical formula would help the readers (and the reviewers) in understanding what was done. Finally, no model metric was reported (AIC, BIC). Were all models performing in a similar way? How well these models explained the observations?

AU: The formula for the logistic regression formula is actually quite straightforward. However we have added it, see line 157. 

The AIC (‘the smaller the better’) is especially used to find the most parsimonious model which was not the goal of the study. The goal of the paper was to rank factories, not to find the best fitting model, however, we deemed it necessary to investigate (and remove) potential confounding of the factory estimates by covariates. 

Finally, I suggest a thorough read of the manuscript before potential resubmissions.

AU: thank you for this advice, we did.

Minor comments

General: the term “factory lesions” sounds odd, it seems like the lesions happened at the factory. I can live with that, but maybe “internal lesions” or something else could be more straightforward for the readers.

AU: factory lesion is a quite common term in TB literature where slaughterhouses are involved (e.g. [4-6]); it means ‘lesions detected in the factory during the slaughter process from antemortem test negative animals’ and is equivalent to Lesions at Routine Slaughter (LRS; e.g. [3,7] which is primarily used in the UK. We have clarified this in line 65 - 66.

Abstract: it would be more beneficial for the readers to have some more background and broader conclusions and considerations in this section, rather than delving into the details of the results.

AU: While we appreciate the reviewer may have read the paper to their taste, we feel that our presentation within the abstract is appropriate for the subject matter and stakeholder audience likely to read or cite this work. Line 14-18 contextualise the work in terms of bovine TB programme within Ireland; line 19-39 outline our approach and major findings; and lines 40-44 present a conclusion. We feel that readers will be informed primarily by the data, more discursive information is presented in the body of the paper for the reader to explore.

Line 49: in Europe only?

AU: the stated reference [2] is about Europe, so we cannot generalise it based on this reference. 

Line 56: remove “thus”.

AU: it has been removed.

Line 58: I know that the presence of bTB deer is still controversial, but it is worth mentioning (see Crispell et al., 2020).

AU: thank you, we added the reference, see line 60.

Line 65: the badger culling program?

AU: no, it is culling of cattle; the SICTT targets are bovines which are eliminated if they turn out to be reactors.

Line 81: why an interval is reported here? Is it the percentage of breakdowns per year?

AU: yes the reviewer is correct, the interval shows the variation in the breakdowns per year, sentence adapted (line 84).

Lines 80-83: would it be possible to report all three data in the same format, like all number or percentages (better) of breakdowns disclosed? Otherwise they are difficult to compare.

AU: unfortunately the percentage cannot be deduced from reference [4]. Percentages are more meaningful than numbers as denominators might change considerably over years.

Line 92: remove “the” in front of Ireland.

AU: it has been removed.

Lines 93-94: confusing, please just report the range.

AU: we adapted the text in line 105 – 106.

Lines 99-105: these sentences are convoluted and hard to read.

AU: we have rephrased the paragraph (lines 112-119). Hopefully it is now more clear.

Line 113: is DOD because of “date of death”?

AU: indeed, it has been replaced to date of death (line 127).

Line 127: slaughtered in the same herd or sent to slaughter from the birth herd?

AU: thank you for the suggestion, it has been replaced (line 141).

Line 137: which distribution was calculated? And where is it reported?

AU: it is the frequency distribution. We have adapted it in the text and the distribution is reported in the result (Table 2).

Table 1: are slaughterhouses Z, ZA, ZB, ZC, ZD, ZE related or there are just not enough letters? Did you consider to switch to number or a code instead (S01, S01 or any)?

AU: thank you, we have recoded the factories S01 to S31 in Table 2.

Table 1 and Figure 1: are the slaughterhouses in random order? Would it be more valuable for the readers if they were ranked according to number of slaughtered animals (or some other metric), instead?

AU: the order in Table 2 is in alphabetic order of the (anonymised) slaughterhouse code in the original dataset and is now ordered by magnitude of submission risk. The order in figure 1 and 2 is based on OR (highest to lowest). 

Table 1: how solid is the inclusion of slaughterhouse T in the study, given only 2 samples submitted to the lab?

AU: indeed, it seems not to be very solid, but factory T (now S31) fulfilled the inclusion criteria (line 173); it also underlines the large variation between factories.

Lines 173-174: confusing, please rephrase.

AU: we have rephrased (line 193-194).

Line 180: “these animals” are the ones with at least one inconclusive test?

AU:. yes the reviewer is correct. They were inconclusive animals.

Line 185: were “higher”?

AU: ”increased” has been replaced with “higher” (line 205).

Line 186: how is the 3-year threshold being selected? Does it make a difference if the 2 or 4 years would be selected to defined the past bTB history? This sensitivity analysis would be material for supplementary information.

AU: data on how many years a herd has been TB free is not available so we cannot categorise it differently.

Line 224: “lower”?

AU: it has been replaced.

Lines 253-261: I struggle to understand the point of this paragraph, in particular 257-261. Why the VOI procedure should make any difference, if it’s not in place in Ireland?

AU: thank you, we have removed line 257 – 261.

Lines 271-275: were these factors addressed in this analysis?

AU: unfortunately, there is no data available for these factors. These factors might explain part of the variation between factories, probably these factors could be investigated in future studies.

Lines 284-287: herd type, breed, sex and age are intimately intertwined factors in modern cattle industry. Since they are mostly treated as stand-alone factors, I would suggest to integrate them at least in the discussion, so to provide the readers a clearer picture.

AU: We concur that these characteristics can be intertwined given developed cattle milk and beef industries. In this sense, we have discussed how there are inter-relationships between these animal level characteristics, though there is enough variation to fit these variables as independent factors without adding large variance inflation. We have now acknowledged this in the discussion, see line 301.

Lines 323-332: this paragraph is a repetition from the introduction, rather than discussion material.

AU: thank you, it has been moved to the introduction section.

S1: I might have missed this in the text, but why some factories have been excluded from the study?

AU: the exclusion criteria are mentioned in line 173-175.

 

Reviewer #2: The manuscript “revisiting the relative effectiveness of slaughterhouses in Ireland to detect tuberculosis lesions in cattle (2014-2018)” updates the analyses carried out in the past regarding the differences between slaughterhouses in Ireland in the rate of submission and laboratory confirmation of tuberculosis-compatible lesions. Given the importance of passive surveillance in the eradication programs for bovine tuberculosis in countries where the disease is at (more or less) reduced levels, the continued evaluation of its performance is an objective worth pursuing, and therefore the topic under study here has scientific merit. The analytical framework used is well established and was used in the past with the same objective, and the interpretation of the results is sound. Still, the addition of some details in terms of methodology and results could help to better understand what was performed, and what the results were.

AU: thank you for the review. We really appreciate it. 

Comments:

- Line 21-22: the term “ever inconclusive or severe inconclusive skin test reactor” is not very straightforward, suggest replacing it by “ever reactor in the standard or severe skin test” or similar (more similar to what was used in lines 30-31)

AU: thank you for the suggestion. It has been replaced (line 21-22). 

- Line 46-47: I would suggest the authors to refer to bovine tuberculosis as the disease caused by members of the M. tuberculosis complex in bovine rather than the disease caused by M. bovis in any host species (though both definitions are widely used in the literature, I think we would not refer to bovine brucellosis as the infection by B. abortus in sheep, and the former definition is more aligned with current regulations in the EU).

AU: we changed it accordingly. Please see line 47-48.

- Line 51: suggest replacing “in” by “by” (since countries did not gain the OTF status precisely in 2019)

AU: replaced (line 52).

- Lines 94-97: Given that in the papers mentioned in the previous lines (refs 15 and 20) at least some of the covariables mentioned here (age, gender…) were considered in the multivariable models, I would expect that variations in their distributions depending on the slaughterhouse should not explain the difference in submission risk per slaughterhouse (that is the point of considering them in the analysis, isn’t it?).

AU: yes, the reviewer is correct, covariates were added in order to get adjusted estimates for the factories and in that way the submission risk does not depend anymore on the differences in type of animals slaughtered.

- Line 113: I think gender is a trait only applicable to humans, and sex should be used here (as e.g. used in Table 1).

AU: corrected (line 108 and 127).

- Line 114: for purchased animals: did the herd history include all the annual tuberculin testing in all herds where the animal had stayed or only the last one prior to be shipped to the abattoir? (the last testing date is mentioned in line 115 but I wonder if this refers to the whole information on skin testing).

AU: this is a good point. The herd history concerns information on the herd the animal last resided in before slaughter. This was a herd level factor, and therefore did not “follow” the animal, and was included for the potentiality of animal level exposure if residual infection was a problem in the herd. ‘Life long’ metrics of exposure would be a useful factor to explore for a future analysis. 

- Data analysis: Herd size is an interesting covariable but I wonder if it makes sense to assume a linear relationship between the (logit of) the probability of submission/confirmation and the number of animals in the herd. Did the authors consider categorizing in some way this variable and see whether it improved (or not) model fit? Although categorizing a continuous variables has its own drawbacks, in terms of interpretation it may be easier to see increase in the risk in “large” vs. “small” herds (just a suggestion). Also, it seems (based on Table S4) it was log-transformed; if so, this should be indicated in the methodology section.

AU: indeed categorising continuous variables should be done on some (bio)logical basis. We ran the models with 4 herd categories based on quartiles. It did not have an effect on factory rankings. Log transformation is now mentioned in M&M (Table 1).

- Data analysis (II): I would be interested in knowing the view of the authors regarding considering factory as a random rather than as a fixed effect. Though I see the point in considering it a fixed effect, given the very large number of categories and the potential objective of characterizing variability across slaughterhouses assuming the belong to some sort of population, coupled with the view of the database as a hierarchical structure (with animals clustered in abattoirs). Note that I don’t think there is anything wrong with the approach used and I am not requesting the authors to change it, but I would like to know if this was considered at some point or if the authors have a strong view against this option.

AU: we did not consider this as an option as we needed to have estimates (odds ratios) in order to rank the factories. In some papers factory is included as random effect but then the question behind the analyses was different, i.e. to get adjusted estimates for specific risk factors, see e.g. [3].

- Line 155: I was surprised to see animals <1 year were excluded but looking at S2 I realize they represent a very small fraction of the population. In other countries it may be not so unusual to slaughter animals below one year (10-11 months-old), so perhaps the fact that animals below 1 year were excluded could be added in the material and methods as an exclusion criterion

AU: thanks for this useful comment, we added it as exclusion criterion, see line 173-175.

- Line 153: suggest to specify that there are four small factories (I think there are four based on line 158, 31 factories after exclusion of small ones).

AU: we have added the number of factories excluded in line 173.

- Lines 161-162: what does this correlation refer to? The crude submission and confirmation risks per factory? This could be perhaps added as a supplementary graph (and although I assume it is considered included in the “descriptive analysis” part on the material and methods, also be specified there as an analysis performed).

AU: indeed it concerns the confirmation risk and submission risk of the factories. A supplementary graph would just show that there is no relation between submission risk and confirmation risk as the correlation is nearly zero. We now have mentioned the correlation in M&M (line 150-151).

- Line 175: suggest replacing “ranged” by “ranging”

AU: it has been replaced (line 195).

- Modeling outcomes: I would suggest to start both submission and confirmation sections by indicating which variables (all considered, I pressume?) were included in the final model. 

AU: we adapted the text. See lines 211-212 and 237-238.

-Also, I would strongly suggest to add a table in the main text with the adjusted OR coefficients (along with some of the descriptives included in Table S2?) for these covariables (something similar to Table 1 in the paper by Frankena et al). Though this is included in table S4, I think it is very useful information that readers would appreciate having in the main text (and could be easily integrated in Table 1, that currently conveys only limited information).

AU: thank you for the useful suggestion. The table with descriptive and OR of risk factor used in the analysis has been added to the text (Table 3).

- Line 191-192 and 216-217: given that factory-specific ORs are calculated based on the difference with the baseline abattoir, I am not sure how to interpret these sentences (differences between factories become larger once covariates were considered? But this would be only considering the reference category, wouldn’t it? Perhaps other differences decrease?)

AU: Thank you very much for this remark. Indeed the difference is relative to the reference category. We adjusted the text to avoid confusion .

- Line 251: suggest replacing “difference” by “increase” (since I think this is what is meant).

AU: replaced (line 275).

- Line 257-261: did the meat inspection procedure change in Ireland at some point though?

AU: thanks for the question, no major changes have been applied.

- Lines 276-287: discussion of the observed effects of age and sex is interesting. Did the authors consider a potential interaction between these covariates?

AU: no, we did not evaluate interactions as the main interest was on the factory ranking.

- Lines 293-295: are these lymph nodes from inconclusive cattle counted as “submitted” in the database?

AU: it depends on the farmer’s decision, sending inconclusive animals to slaughter is not compulsory. A very high % from the ever inconclusive animals is submitted, regardless of the presence of visible lesions. This then drives down the proportion that are confirmed. The opposite is happening with the ever severe inconclusive, where a relatively small proportion is submitted with a very high confirmation %.

I guess this is an important source of error since for the rest submission will always be linked to the presence of lesions while it wouldn’t be the case for the inconclusive cattle. I would suggest to declare this more openly as a limitation of the study that – hopefully – affects a minor proportion of animals in the database (less than 20k animals in the whole database). Would these animals be preferentially submitted to certain slaughterhouse?

AU: thanks for this useful comment. Indeed, it is a minor proportion of the ever inconclusive cattle (11,526/6,578,245 =0.17%, Table 3). We referred to this in the Discussion, see lines 316-320. Further analysis of the data show that many factories process ever inconclusive animals. Thus, these animals are not processed in specific slaughterhouses.

- Lines 302-305: this is a very long sentence that doesn’t read very easily, suggest to break it in two sentences and revise it.

AU: we rephrased the sentence (328-332).

- Lines 323-332: this seems more an introductory section, I would suggest to remove it (or relocate it in the introduction).

AU: thank you, we have relocated the paragraph to the introduction section, see lines 87 – 95.

- Are (most of) the factories the same as in the previous papers by Frankena et al and Olea-Popelka et al.? If so, and is possible for the authors, it would be interesting to comment on whether factories remain in the same ranking (or quartile) over time.

AU: factories are partly the same but as all were anonymised in all studies no comparison can be made; only the average levels and the variation between factories can be evaluated.

- A minor suggestion is to remove the second decimal in numbers above 10% (e.g., replace 40.72% by 40.7%, since with large numbers it seems unnecessary). 

AU: in principal the reviewer is correct. However, for consistency we used two decimals for all small and large figures.

Also, I think sentences should not start with a number (e.g., line 159, start with e.g. “A total of 13,337” instead of by “13,337” directly after the full stop).

AU: thank you very much for the useful suggestion, we adapted the text where it occurred.

1. Frankena K, White PW, O'Keeffe J, Costello E, Martin SW, Van Grevenhof I, et al. Quantification of the relative efficiency of factory surveillance in the disclosure of tuberculosis lesions in attested Irish cattle. Vet. Rec. 2007;161(20):679-84. doi: 10.1136/vr.161.20.679.

2. Olea-Popelka FJ, Freeman Z, White P, Costello E, O'Keeffe J, Frankena K, et al. Relative effectiveness of Irish factories in the surveillance of slaughtered cattle for visible lesions of tuberculosis, 2005-2007. Ir. Vet. J. 2012;65(1). doi: 10.1186/2046-0481-65-2.

3. Pascual-Linaza AV, Gordon AW, Stringer LA, Menzies FD. Efficiency of slaughterhouse surveillance for the detection of bovine tuberculosis in cattle in Northern Ireland. Epidemiol. Infect. 2017;145(5):995-1005. doi: 10.1017/S0950268816003095.

4. Byrne AW, Barrett D, Breslin P, Madden JM, O’Keeffe J, Ryan E. Post-mortem surveillance of bovine tuberculosis in Ireland: herd-level variation in the probability of herds disclosed with lesions at routine slaughter to have skin test reactors at follow-up test. Vet. Res. Commun. 2020;44(3-4):131-136. doi: 10.1007/s11259-020-09777-w.

5. Willeberg PW, McAloon CG, Houtsma E, Higgins I, Clegg TA, More SJ. The herd-level sensitivity of abattoir surveillance for bovine tuberculosis: simulating the effects of current and potentially modified meat inspection procedures in Irish cattle. Front. Vet. Sci. 2018;5(82). doi: 10.3389/fvets.2018.00082.

6. Clegg TA, Good M, More SJ. Future risk of bovine tuberculosis recurrence among higher risk herds in Ireland. Prev. Vet. Med. 2015;118(71-79). doi: 10.1016/j.prevetmed.2014.11.013.

7. Shittu A, Clifton-Hadley RS, Ely ER, Upton PU, Downs SH. Factors associated with bovine tuberculosis confirmation rates in suspect lesions found in cattle at routine slaughter in Great Britain, 2003–2008. Prev. Vet. Med. 2013;110(3-4):395-404. doi: 10.1016/j.prevetmed.2013.03.001.

---

## [Decision Letter · Decision Letter 1]

7 Sep 2022

PONE-D-22-10323R1Revisiting the relative effectiveness of slaughterhouses in Ireland to detect tuberculosis lesions in cattle (2014-2018)PLOS ONE

Dear Dr. Male Here,

Thank you for submitting your manuscript to PLOS ONE. After careful consideration, we feel that it has merit but does not fully meet PLOS ONE’s publication criteria as it currently stands. Therefore, we invite you to submit a revised version of the manuscript that addresses the points raised during the review process.

Reviewer #2 has raised one point that should be clarified.

We look forward to receiving your revised manuscript.

Kind regards,

Rebecca Lee Smith, D.V.M., M.S., Ph.D.

Academic Editor

PLOS ONE

Journal Requirements:

Reviewers' comments:

Reviewer's Responses to Questions

**Comments to the Author**

1. If the authors have adequately addressed your comments raised in a previous round of review and you feel that this manuscript is now acceptable for publication, you may indicate that here to bypass the “Comments to the Author” section, enter your conflict of interest statement in the “Confidential to Editor” section, and submit your "Accept" recommendation.

Reviewer #1: (No Response)

Reviewer #2: All comments have been addressed

2. Is the manuscript technically sound, and do the data support the conclusions?

Reviewer #1: Partly

Reviewer #2: Yes

3. Has the statistical analysis been performed appropriately and rigorously? 

Reviewer #1: Yes

Reviewer #2: Yes

4. Have the authors made all data underlying the findings in their manuscript fully available?

Reviewer #1: No

Reviewer #2: Yes

5. Is the manuscript presented in an intelligible fashion and written in standard English?

Reviewer #1: Yes

Reviewer #2: Yes

6. Review Comments to the Author

Reviewer #1: The effort of the authors made in addressing the syntax and clarity of the manuscript were sound, and the methodology and model explanation are cleaer now. However, they were short in addressing many others concerning aspects concerning terminology (which could have been improved, but it wasn't) and, in particular, giving a broader context and depth to the results, and making the study a more interesting one.

Reviewer #2: The revised version of the manuscript has satisfactorily addressed the points raised in the previous review. I have only a question and a couple of editorial suggestions

- I was a bit confused by one of the replies given to reviewer#1 regarding the purpose of multivariable models: this reply stated “in the univariable model only Factory_id was included as fixed effect (i.e. all factories) and herd_id as random effect”. From table 3 it seems obvious that in fact all available covariates were expressed in univariable models (this makes sense in my view). Based on the answer I assume that these univariable models also included a random effect (I think this also makes sense), and I assumed they did not include the factory (but based on the answer I am not so sure anymore). Perhaps this could be better clarified in lines 154-155 (version with track changes). A suggestion would be “The effect of each risk factor on the FL submission and confirmation risk was analysed with multilevel univariable logistic regression, followed by multilevel mixed effects logistic regression with…”, i.e., adding the multilevel to the univariable part to make evident that random effects were also included in the univariable models. Then, if all univariable models included the factory, perhaps state in that sentence this (since if the model includes e.g. factory and year does not fit the classical definition of a univariable regression model).

- Lines 234 and 260 (version in track changes): in the captions of figures 3 and 4, replace “referent” by “reference category”

7. PLOS authors have the option to publish the peer review history of their article (what does this mean?). If published, this will include your full peer review and any attached files.

Reviewer #1: No

Reviewer #2: No

---

## [Author Response · Author response to Decision Letter 1]

9 Sep 2022

We appreciate very much the time and effort the reviewers have taken to read and comment on our manuscript. The comments and suggestion were very helpful to improve the paper and below you can find all our responses (in bold italics) with indications to where we have changed the manuscript. We hope that our answers to the queries are satisfactory and that the revised version is now acceptable for publication.

On behalf of all authors,

Rischi Robinson Male Here

(corresponding author)

Reviewer #1: The effort of the authors made in addressing the syntax and clarity of the manuscript were sound, and the methodology and model explanation are cleaer now. However, they were short in addressing many others concerning aspects concerning terminology (which could have been improved, but it wasn't) and, in particular, giving a broader context and depth to the results, and making the study a more interesting one.

AU: Thank you for your time and consideration. The terminology is in accordance with previous publications on the same topic both from the same group (Frankena, Olea-Popelka) as from other groups (Northern Ireland). Consistency indeed is important and we have the opinion to have done so.

Reviewer #2: The revised version of the manuscript has satisfactorily addressed the points raised in the previous review. I have only a question and a couple of editorial suggestions

- I was a bit confused by one of the replies given to reviewer#1 regarding the purpose of multivariable models: this reply stated “in the univariable model only Factory_id was included as fixed effect (i.e. all factories) and herd_id as random effect”. From table 3 it seems obvious that in fact all available covariates were expressed in univariable models (this makes sense in my view). Based on the answer I assume that these univariable models also included a random effect (I think this also makes sense), and I assumed they did not include the factory (but based on the answer I am not so sure anymore). Perhaps this could be better clarified in lines 154-155 (version with track changes). A suggestion would be “The effect of each risk factor on the FL submission and confirmation risk was analysed with multilevel univariable logistic regression, followed by multilevel mixed effects logistic regression with…”, i.e., adding the multilevel to the univariable part to make evident that random effects were also included in the univariable models. Then, if all univariable models included the factory, perhaps state in that sentence this (since if the model includes e.g. factory and year does not fit the classical definition of a univariable regression model).

AU: Thank you for your comment and suggestion. We apologize and we understand your confusion. The reviewer is correct that all available covariates were expressed in univariable models as shown in Table 3. Also, we want to clarify that factory was not included in these univariable models. In the first rebuttal letter it was stated that herd was included as random effect in the univariable models; however the OR’s and CI’s in Table 3 originated from univariable models without that random effect. Therefore we corrected the univariable results presented in Table 3 and updated the text and figures where necessary, leading to very minor changes. Also we now explicitly mention in M&M that the univariable models included a random effect (line 152).

Lines 234 and 260 (version in track changes): in the captions of figures 3 and 4, replace “referent” by “reference category”.

AU: Thank you for the suggestion. It has been replaced (line 230 and 255).

To the editor: we would like to update the funding disclosure to clarify the funder of the article processing costs. The updated funding disclosure is given below: 

“The article processing costs associated with the publication of this study were provided by the Department of Agriculture, Food and Marine, Ireland. RRMH was sponsored by Indonesia Endowment Fund for Education scholarship (LPDP, https://lpdp.kemenkeu.go.id) from the Ministry of Finance, the Republic of Indonesia. The funder had no role in study design, data collection, and analysis, decision to publish, or preparation of the manuscript”.

---

## [Editor Report · Decision Letter 2]

13 Sep 2022

Revisiting the relative effectiveness of slaughterhouses in Ireland to detect tuberculosis lesions in cattle (2014-2018)

PONE-D-22-10323R2

Dear Dr. Male Here,

We’re pleased to inform you that your manuscript has been judged scientifically suitable for publication and will be formally accepted for publication once it meets all outstanding technical requirements.

Kind regards,

Rebecca Lee Smith, D.V.M., M.S., Ph.D.

Academic Editor

PLOS ONE
---

## [Editor Report · Acceptance letter]

28 Sep 2022

PONE-D-22-10323R2 

Revisiting the relative effectiveness of slaughterhouses in Ireland to detect tuberculosis lesions in cattle (2014-2018) 

Dear Dr. Male Here:

I'm pleased to inform you that your manuscript has been deemed suitable for publication in PLOS ONE. Congratulations! Your manuscript is now with our production department. 

Kind regards, 

on behalf of

Dr. Rebecca Lee Smith 

Academic Editor

PLOS ONE